# Investigation of Morphological, Chemical, and Thermal Properties of Biodegradable Food Packaging Films Synthesised by Direct Utilisation of Cassava (*Monihot esculanta*) Bagasse

**DOI:** 10.3390/polym15030767

**Published:** 2023-02-02

**Authors:** Vindya Thathsaranee Weligama Thuppahige, Lalehvash Moghaddam, Zachary G. Welsh, Azharul Karim

**Affiliations:** 1School of Mechanical, Medical and Process Engineering, Queensland University of Technology, Brisbane, QLD 4001, Australia; 2Centre for Agriculture and Bioeconomy, Queensland University of Technology, Brisbane, QLD 4001, Australia; 3Department of Food Science and Technology, Faculty of Agriculture, University of Ruhuna, Mapalana, Kamburupitiya 81100, Sri Lanka; 4School of Chemistry and Physics, Queensland University of Technology, Brisbane, QLD 4001, Australia

**Keywords:** agro-industrial waste, biodegradable food packaging film, characterisation, eco-friendly process, hot-pressing, powdered cassava bagasse, synthesis

## Abstract

The utilisation of edible sources of starch such as corn, wheat, potato, and cassava has become the common approach to develop biodegradable food packaging. However, the future food security issue from the wide application of such edible starch sources has become a major concern. Consequently, exploring non-edible sources of starch for starch-based biodegradable food packaging and their property enhancement have become one of the common research interests. Although there has been a great potentials of synthesising biodegradable food packaging by direct utilisation of agro-industrial waste cassava bagasse, there have been very limited studies on this. In this context, the current study investigated the potential of developing biodegradable food packaging by directly using cassava bagasse as an alternative matrix. Two film-forming mixtures were prepared by incorporating glycerol (30% and 35%), powdered cassava bagasse and water. The films were hot-pressed at 60 °C, 100 °C, and 140 °C temperatures under 0.28 t pressure for 6 min. The best film-forming mixture and temperature combination was further tested with 0.42 t and 0.84 t pressures, followed by analysing their morphology, functional group availability and the thermal stability. Accordingly, application of 35% glycerol, with 100 °C, 0.42 t temperature and pressure, respectively, were found to be promising for film preparation. The absence of starch agglomerates in film surfaces with less defects suggested satisfactory dispersion and compatibility of starch granules and glycerol. The film prepared under 0.42 t exhibited slightly higher thermal stability. Synthesised prototypes of food packaging and the obtained characterisation results demonstrated the high feasibility of direct utilisation of cassava bagasse as an alternative, non-edible matrix to synthesise biodegradable food packaging.

## 1. Introduction

In modern days, food packaging has become one of the most essential means of presenting food items to the consumers. While playing an important role in containing the food item in storage or during transportation, the package also aims to preserve the quality of the packed food [1]. It also indirectly reduces food waste and ensures food security by enhancing the shelf life of the food product [2]. Additionally, food packaging delivers the information on the packed food to the consumers and indirectly caters as a powerful marketing tool by influencing consumer behaviour [3]. Moreover, the consumers’ expectation of eco-friendly food packaging with higher food integrity and advanced functionalities, such as smart and active packaging over conventional food packaging, is becoming stronger due to the growing concerns of consumers on environmental sustainability and health consciousness [3,4]. In order to have the maximum consumer satisfaction to the packed products, such sustainable and effective food packaging, along with other sophisticated features, are of a huge interest of the food manufacturers. Although the conventional food packaging possesses superior packaging properties (i.e., mass transfer, mechanical and barrier properties), the majority of them are made from petrochemical-based materials such as polyethylene terephthalate (PET), poly (trimethylene terephthalate) (PTT), polypropylene (PP), polyvinyl chloride (PVC), and ethylene vinyl chloride (EVOH) [5], which are non-biodegradable and problematic to the environment. Moreover, the higher the use of such conventional food packaging, the higher the pressure on the non-renewable petroleum sources [6]. Therefore, at present, consumers’ perception on environmental sustainability has significantly strengthened the importance of developing biodegradable food packaging, investigating their properties and functionalities as an alternative to conventional food packaging.

The ability of complex organic materials to be decayed into simplified substances as a result of microorganism activities is defined as biodegradability [7]. Polymers (either natural or synthesised) that possess the ability to break down into natural by-products (i.e., gases, water, biomass, etc.) after their intended purpose are known as biodegradable polymers [8]. Polymer degradation is governed by several material-related and environmental related factors. Importantly, material-wise, the chemical structure of the material including functional groups available and their stability are amongst the major factors affecting the biodegradability of a particular material, whereas physical properties, morphology and physico-mechanical properties are also playing important roles [8]. Degradation generally causes significant alterations in the polymer properties compared to that of their initial properties. Therefore, maintaining the intended purpose-related properties without compromising the biodegradability of polymers are always challenging specifically in the biodegradable food packaging sector. To date, various biopolymers such as different starch [9,10,11,12,13], chitosan [14,15,16], protein [17,18], gelatine [19], bacterial cellulose [20,21], etc., have been comprehensively studied to develop biodegradable food packaging materials. Based on their origin, these biopolymers have been categorised as bio-technically synthesised (i.e., Polylactides), microorganism derived (i.e., polyhydroxyalkanoates/PHAs, poly(3hydroxybutyrate-co-3-hydroxyvalerate/PHBV), and biomass derived biopolymers (i.e., polysaccharides and proteins) [5]. Amongst them, starch, a commonly found biomass-derived biopolymer, has been widely studied as a matrix for biodegradable food packaging. Irrespective of the starch sources, starch-based food packaging possesses poor food packaging properties and functionalities [22,23]. This major drawback has prevented its widespread application over conventional food packaging. Therefore, continuous investigations to explore ideal candidates for starch-based biodegradable food packaging matrices and enhancing their food packaging properties and functionalities are of crucial importance. Consequently, many research works have been conducted utilising different sources of starch, along with various polymers, plasticisers, cross-linkers, compatibilisers, and biocomposites to develop better biodegradable food packaging. However, the majority of starch sources investigated can be classified as edible sources (e.g., corn, wheat, potato, sweet potato, and cassava), which could create additional food security concerns if starch-based biodegradable food packaging is widely adopted. Alternative, non-edible sources of starch are worthwhile to investigate for the biodegradable food packaging sector to minimise food security concerns. Application of agro-industrial waste as alternative biodegradable food packaging matrices would ensure better food security, assure eco-friendliness and help achieving economic benefits through their valorisation [24]. Cassava (Manihot esculenta) bagasse or cassava pulp is a cassava industrial waste and has been reported to be an excellent source of residual starch and fibre [25]. Generally, agro-industrial waste cassava bagasse consists of moisture (8.70%), ash (2.15%), protein (1.65%), reducing sugar (0.12), glucose (0.02%), and significantly high residual starch content (47.75%) [26]. Except for the recent research endeavours by Silviana, Brandon and Silawanda [27] and Silviana and Dzulkarom [28] to develop a biodegradable film using cassava bagasse starch with chicken feet gelatine and develop a cassava bagasse starch-based film incorporated with bamboo fibre, respectively, the utilisation of cassava bagasse as an alternative starch matrix to the synthesis of biodegradable food packaging has not been reported so far. Instead, their applications in the biodegradable food packaging sector as sources of fibre to obtain reinforced mechanical properties have been frequently studied [29,30,31,32]. However, the potential of direct utilisation of cassava bagasse to be used as both starch and fibre rich alternative matrices to synthesis eco-friendly biodegradable food packaging has not been a popular concept so far. Reduced efforts, cost, and time to extract starch and fibre from cassava bagasse while valorising agro-industrial waste in a sustainable manner are amongst some of the significant potential benefits of adopting the proposed concept. Moreover, the direct utilisation of such agro-industrial wastes to develop biodegradable food packaging would help to reduce the amount of unused fibrous waste generated through extraction processes.

This study aims to synthesis and characterise a cassava bagasse-based food packaging material in order to assess the feasibility of cassava bagasse as an alternative matrix. Specifically, the current research study will directly use cassava bagasse, by avoiding any additional or separate incorporation of starch (as a starch matrix) or fibrous materials (as reinforced agents) during film synthesis. These attempts on direct utilisation of cassava bagasse as a fibre starch matrix without extracting starch for film preparation would also expect to avoid any constrains associated with its starch extraction process. It is expected that the synthesised films not only possess biodegradability but will also pave the way for the future works to assess the novel films’ packaging properties and functionalities.

## 2. Materials and Methods

### 2.1. Materials

The well-cleaned and sun-dried cassava bagasse obtained from a cassava starch industry based in Thailand was supplied by the Excellent Centre of Waste Utilisation and Management (ECoWaste), King Mongkut’s University of Technology Thonburi (KMUTT), Thailand. Prior to the film synthesis process, the cassava bagasse samples were milled using a planetary mono mill (Fritsch, Pulverisette 6 classic line, Birkenfeld, Germany) at 300 rpm for 3 min. Later, the obtained powdered cassava bagasse was passed through a 425 µm sieve. The powder obtained after the sieving was then stored in a sealed container and kept at room temperature (22 ± 2 °C) until the film preparation was performed.

### 2.2. Film Preparation

The powdered cassava bagasse and glycerol were used as the matrix and the plasticiser, respectively, for the film preparation and no other sources of fibres were added as the reinforcing agents. The methods followed for the preparation of film-forming mixtures and the cassava bagasse-based biodegradable food packaging films are described below. The prepared films were stored at 50% RH and 22 ± 2 °C until the characterisations were performed.

#### 2.2.1. Preparation of Film-Forming Mixture

As shown in Table 1, two film-forming mixtures were prepared by incorporating various proportions of powdered cassava bagasse, glycerol, and water. Then, the mixtures were continuously stirred using a hot-plate magnetic stirrer (Model RCT basic, IKA, Breisgau-Hochschwarzwald, Germany) for 30 min. The speed and the temperature for the stirring process were set as 300 rpm and 90 °C, respectively. Finally, well-homogenised film-forming mixtures were separately collected into air-tight containers and refrigerated until the film-forming was performed. The study conducted by Lai, Don, and Huang [33] for preparing hot-pressed corn starch-based films was used as the initial experimental conditions. The conditions were applied after trial and error.

#### 2.2.2. Synthesis of Cassava Bagasse-Based Biodegradable Food Packaging Film

Approximately 50 mg of prepared film-forming mixtures were placed at a time on the hot-press instrument (YLJ-HP60-LD, USA). The film preparation followed was a two-stage process. In the first stage, the best film-forming mixture and its temperature has been determined based on their macroscopic view. Both the G 30 and G 35 film-forming mixtures were hot-pressed by using three different temperature levels (60 °C, 100 °C, and 140 °C). The constant pressure of 0.28 t was applied during the hot-pressing process at this stage. The optimised film-forming mixture and the temperature combination for successful film preparation were determined upon the macroscopic view. At the second stage, those selected conditions for the hot-press film preparation were again investigated for another two different pressure levels (0.42 t and 0.84 t). These studied pressure levels (0.28 t, 0.42 t and 0.84 t) are representing the conversions of the pressure shown in the instrument as 2 MPa, 3 MPa and 6 MPa, respectively. Since the pressure differences have been identified as a major factor to the characteristics of synthesised films using hot-press instrument [34], three different pressure levels have been studied. Thereby, the two-fold and three-fold enhancement of the initial pressure have been tested for the film preparations during the current study. The duration for each hot-pressing process was 6 min. After 6 min, the successfully prepared hot-pressed thin films were separated from the instrument. Finally, they were stored at room temperature until further characterisations were performed.

### 2.3. Film Characterisation

The hot-pressed films were first investigated by visual observation. These observations were used as a simple criterion to optimise the film-forming mixtures and its related conditions during the hot-press film preparation process. The successfully prepared hot-pressed films were then further characterised for their morphological, thermal, and chemical properties through Scanning Electron Microscope (SEM), Thermogravimetric Analysis (TGA) and Fourier Transform Infrared (FTIR) analysis, respectively.

#### 2.3.1. Macroscopic Characteristics

The films were detached from the hot-press instrument, and their macroscopic characteristics were observed through visual observation. The colour, shape, and the presence of any visible defects such as cracks or grooves within their film surfaces were observed for comparative purposes. Synthesised films which were observed to be fragile during the manual tests or had serious cracks were excluded from further investigations and characterisation while the films with satisfactory film-forming ability have been used for the further investigations.

#### 2.3.2. SEM Analysis

Selected hot-pressed film samples were air dried, mounted on aluminium stubs using double-side tapes and coated with a 4 nm Pt layer. Later, the surface morphology of the hot-pressed films was observed through a scanning electron microscope (Tescan Mira 3 XMU, Brno, Czech Republic). The secondary electron detector was used and 5 kV of applied accelerating voltage was used for the analysis. The SEM micrographs obtained at different magnification levels were used to comparatively investigate the surface morphology of different films.

#### 2.3.3. Fourier Transform Infrared (FTIR) Analysis

A FTIR analysis was performed for the different hot-pressed films using a Nicolet iS50 FTIR spectrometer (Thermo Scientific, Madison, WI, USA) equipped with a smart endurance single bounce diamond ATR accessory. The spectra were collected within the range of 600 to 3800 cm^−1^ by performing sixty-four scans for each film sample. The scanning time and the resolution were set as 60 s and 4 cm^−1^, respectively. The band assignment of the obtained FTIR spectra for the films were used to interpret the functional groups available within the studied films. The obtained data were used to conduct a comparative analysis of chemical characteristics of the prepared film in order to have an understanding on whether there is any significant influence on the chemical composition of the films due to the different applied pressures.

#### 2.3.4. Thermogravimetric Analysis

The thermal stability of the hot-pressed films was determined through a TGA analysis with respect to their resulted thermal decomposition patterns. The analysis was performed by using a Netzsch STA 449F3 Jupiter Simultaneous Thermal Analyser (Netzsch, Selb, Germany) within a temperature range of 30–1000 °C. The heating rate was set as 10 °C/min. Nitrogen was used as the purge gas atmosphere with a flow rate of 50 mL/min during the analysis. The obtained data for the TGA analysis were analysed using the Netzsch Proteus software (Version 8.0.2).

## 3. Results and Discussion

### 3.1. Macroscopic View

The macroscopic views of the hot-pressed films prepared at stage 1 and 2 are shown in Figure 1 and Figure 2, respectively. Figure 1 presents the macroscopic views of the films prepared using both film-forming mixtures (G 30 and G 35). The non-toxic and non-irritating nature of the glycerol with its biodegradable, recyclable nature, and the high stability on typical storage conditions [35] have made it an ideal candidate in terms of developing packaging materials which would ensure the food integrity of the packed food. Notably, Cheng, Chen, McClements, Yang, Zhang, Ren, Miao, Tian and Jin [36] have reported that glycerol possesses the ability of weakening the internal hydrogen bond forces by loosening the structure when incorporated with starchy materials. However, the plasticising effects depends on glycerol concentration. As per the literature, the minimum and maximum requirements of glycerol concentrations were reported to be 30% and 50%, respectively, in order to avoid formation of any ductile or over plasticised films. Research-based evidence provided in the literature that 33% glycerol incorporation into starch matrix has satisfied the requirements of biodegradability [37]. This was the basis for the current study: to incorporate 30% and 35% of glycerol contents into cassava bagasse matrices. Initially, the hot-pressing process was conducted using 30% and 35% of glycerol contents at three different temperature conditions (60 °C, 100 °C, and 140 °C), and their macroscopic views are shown in Figure 1 (G 30 a), (G 30 b), and (G 30 c), respectively. The visual observation reveals that the none of the applied temperatures on G30 were successful in terms of film forming. As shown in Figure 1 (G 30 a) and (G 30 b), cracks were observed on the surfaces of the films. Moreover, as depicted in Figure 1, they were highly fragile and were broken into pieces even with manual handling. Figure 1 (G 30 c) presents the outcome of the application of 140 °C temperature on G 30. As per the results, instead of any film formation, the material was observed to be stuck to the aluminium foil of the hot-press instrument. In addition, a strong caramelised odour was produced during this processing.

Figure 1 (G 35 a) shows the obtained material by applying a 60 °C of temperature on G35 during the hot-pressing process. Instead of exhibiting any film-forming ability, G 35 A observed to have many defects, resulting in high fragility. These observations were also in accordance with the observations for Figure 1 (G 30 a) and (G 30 b). Accordingly, all the prepared materials were found to be fragile and non-flexible. However, these results are providing potential research insights to improvements to the materials and the process towards developing biodegradable materials, which might be applicable to reusable utensils and containers with biodegradability. Amongst the obtained results, only G 35 subjected to the hot-press process at 100 °C temperature (Figure 1 (G 35 b)) was observed to be in the form of a film. Table 1 presents the G 35 film-forming mixture consisting of 35% (weight percentage) of glycerol. Except for slight damages at the edges, no prominent cracks were found on the film surface. Notably, the film was found to be flexible during its manual handling with compared to the rest of the other prepared films shown in Figure 1 (G 30 a), (G 30 b), and (G 35 a).

These results suggest an influence of glycerol on plasticising of cassava bagasse-based hot-press film preparation. According to Hancock and Zografi [38], water plays an important role as a plasticiser for amorphous materials. Cassava bagasse consists of significantly high residual starch content, and the starch consists of a fraction of amorphous materials. Therefore, it is proven that not only glycerol but also water is acting as a plasticising agent. As presented in Table 1, in addition to the increment of 16.67% (weight percentage) of glycerol, G 35 has also accounted for an increment of 21.74% water content compared to that of G 30. As a result, the total weight percentage of glycerol and the water in G 30 and G 35 account for a weight percentage of 53% and 63%, respectively. Thereby, it can be claimed that both the water and the glycerol might have resulted in the comparatively high film formability exhibited by G 35. However, it is noteworthy to mention that only the G 35 at 100 °C during the hot-press process has exhibited the film-forming ability. This suggests that the applied temperature greatly influences on the film-formation ability during the cassava bagasse-based film synthesis. In the presence of plasticisers and water, starch subjects to gelatinisation at around 60–90 °C [39]. The thermoplasticisation of starch occurred during this process due to the molecular structure alterations is beneficial in terms of film formation. All these factors are combinedly influencing the film-formation under the studied conditions. Thereby, this research-based evidence is better explaining the use of G 35 at 100 °C for successful film preparation.

Hot-pressing at 140 °C temperature even with the G 35 has proven to be unsuccessful. The excessive gelatinisation that occured due to the overheating during hot-pressing at 140 °C might cause this failure. According to The BC Cook Articulation Committee (2015), evaporation of water and shrinkages of the gel (i.e., glycerol) are amongst the unfavourable outcomes of excessive gelatinisation. The photographs of films obtained by applying 140 °C in both the Figure 1 (G 30 c) and (G 35 c) also confirm these explanations. Both the figures present an uneven distribution of the solid substances that are non-detachable from the hot-press surfaces. Additionally, a caramelised odour was also produced during the hot-pressing process of the G 35 at 140 °C. In general, the caramelisation odour is uniquely produced during the caramelisation reactions. This process is known as one of the nonenzymatic browning reactions undergone by the sugars at higher temperatures, and, in turn, it results a unique caramelisation odour, brownish colour and alterations in flavours [40]. Starch-based biodegradable films have also reported to be undergone colour changes due to the caramelisation process [41]. According to Ketiku and Oyenuga [42], total sugars (i.e., glucose, fructose, maltose, and sucrose) are found on cassava roots, and their amounts were reported to be fluctuating upon the plants’ maturity stages. Therefore, it can be explained that the experienced caramelisation odour has resulted from the caramelisation reaction. In addition to the stronger odour, the comparatively dark brownish colour was observed within the G 30 over G 35 at 140 °C temperature. Therefore, it can also be suggested that the comparatively higher availability of total sugar within the G 30 could have contributed to significant caramelisation, resulting in significant colour variations. This also suggests the comparatively higher availability of total sugar within the G 30. This is well-depicted through the formulations for film-forming mixture in Table 1. Though it is possible to control these reaction rates, the evidence of the excessive gelatinisation at 140 °C confirms the fact that 140 °C is not an ideal temperature as a hot-pressing condition for producing cassava bagasse-based biodegradable food packaging materials.

Moreover, the unique caramelisation odour and the colour produced during the process might not be beneficial as appliances for food packaging materials. Generally, the available high sugar content and high temperatures are favourable for the caramelisation process. Therefore, this process can also expect to occur even at 100 °C during film synthesis, while significantly affecting the quality of the final film. To an extent, the caramelisation odour imparting to the film might be a major concern for the food integrity of the packed products. Moreover, the dark-coloured films might contribute to reduce film transparency/light distraction [41], which might trigger the consumers’ behaviour while purchasing a product within such packaging. Since maintaining food integrity and consumer perception are two important aspects of the quality of any packaging, a necessity is emerging to explore the necessity of pre-treating the matrices to remove such sugar, and to investigate the effects of such processes on the film synthesis process.

Notably, variation in the colour has been observed for the hot-pressed films at different glycerol contents and hot-pressing temperatures. Although the colour variation of G 30 A, G 35A, and G 30 B were observed to be insignificant, the colour variation of synthesised G 35 B films observed under the naked eye has been comparatively high. It was reported that the higher the plasticisation effect results in higher polyol content which imparts colour to the films [37]. Accordingly, the film-forming ability exhibited by G 35 B also confirms that the comparatively excellent plasticisation effect occurred during the synthesis of G 35 B. Based on these observations, the film formed with the G 35, at the temperature, pressure, and hot-pressing time of 100 °C, 0.28 t and 6 min, respectively, was selected for further investigations for two different pressure levels and their characterisation. Figure 2a,b present the photographs of the films prepared with an applied pressure of 0.42 t and 0.84 t, respectively. Similar to the film shown in Figure 1 (G 35 B), the edges of both these films were also observed to be slightly damaged. Both films have no significant cracks and were flexible during their manual handling, which was a qualitative measurement. However, the quantitative measurements of the flexibility of each synthesised prototypes of cassava bagasse-based biodegradable food packaging films will only be revealed through carrying out the elongation of break and/or puncture tests. Overall, the macroscopic characteristics of the three films prepared using the G 35 at 100 °C under three different pressure levels did not exhibit any significant deviations from each other.

### 3.2. Surface Morphology of the Films

A SEM analysis was conducted to investigate the surface morphology of three different hot-pressed films prepared using G 35 at 100 °C. The obtained micrographs of the films with three applied pressure levels (i.e., 0.28 t, 0.42 t and 0.84 t) are shown in Figure 3a–c, respectively. As can be seen, all the films reveal an approximately similar surface morphology. Instead of being ideally smooth, as shown in Figure 2, the film surfaces were observed to consist of (1) some non-melted starch granules, (2) a few voids, and (3) a few cracks. Similar defects were also reported by Amin, Chowdhury and Kowser [22] and Reis, Pereira, Smith, Carvalho, Wellner and Yakimets [43] on the surfaces of prepared starch/composite bioplastics with corn starch matrix and polyhydroxybutyrate-hydroxyvalerate (PHB-HV)/maize starch, respectively. As shown in Figure 3a–c, the distribution of these defects is not continuous on the surfaces, whereas they appear less frequently. Although non-melted starch granules were randomly observed, any starch agglomerates were absent on the film surfaces. These results suggest that the starch granules were well dispersed during the film-forming process. The obtained results are in contrast with the results reported by Reis, Pereira, Smith, Carvalho, Wellner and Yakimets [43] for an incompatible blend of PHB-HV/maize starch. As reported by Jost and Langowski [44], causes for such structural defects is not only limited to the actual changes occurred during the film processing but also might be due to the evaporation of the plasticisers. Moreover, it is possible that the small cracks might have propagated during the drying process of the films.

Righetti, Cinelli, Mallegni, Stabler and Lazzeri [45] have reported that the structural defects on the film surfaces affect their mechanical properties. However, Versino, López and García [46] have reported that the incorporation of cassava bagasse has affected the morphology of the thermoplasticised cassava starch films, which has been reported to be beneficial in terms of their mechanical properties. According to Righetti, Cinelli, Mallegni, Stabler and Lazzeri [45], absence of any pulled-out fibrous particles on film surfaces suggests the satisfactory adhesion of the fibre to the film matrix. The findings of the current study were also in accordance with their claim and suggested the satisfactory adhesion of the fibre to the starch in the cassava bagasse matrix. Based on these observations, although none of the obtained film surfaces were ideally smooth and homogenous, cassava bagasse has demonstrated an impressive film-forming ability despite being a non-starchy matrix. The discussed plasticising effect of the G 35 film synthesised at 100 °C based on its colour variation in the previous section is also well-explained through the observed SEM micrographs.

However, previous studies have reported that the non-melted starch granules within the surface of the synthesised films are a good indicator of the lack of compactness and the homogeneity of the film matrix, which might also result in weak mechanical properties compared to the surfaces with high structural integrity [47]. Moreover, the functional properties (water vapour, oxygen permeability, wettability, transparency, mechanical properties) of starch-based biodegradable films can also be affected by the amount of incorporated glycerol, due to the potential of forming a looser structure compared to the films synthesised without glycerol [36]. Moreover, the higher glycerol content has reported to contribute to the transparency and higher hydrophilicity of the starch-based films with reduced tensile strength [37,48]. Therefore, availability of such non-soluble starch granules within the synthesised film surfaces along with the glycerol incorporation can affect the functional properties of the synthesised films by acting as major factors that contribute to the less compactness and heterogeneity of the film surfaces, which have resulted in physical defects. Moreover, although the structural defects such as cracks have also reported to be indicators of poor bonding between the constituents within film-forming blends [22], a small number of non-prominent defects observed during the current study suggest a satisfactory compatibility of cassava bagasse and glycerol, which prove the potential of this novel concept. Additionally, although the literature suggests that the incorporation of high glycerol concentration may affect the functionalities of the films, it is expected that high glycerol content would possess beneficial biodegradability to the films [37]. However, since the film-forming solution (G 35) is only formulated with powdered cassava bagasse, glycerol, and water, its modifications with various crosslinkers and compatibilisers is recommended to enhance the compactness and the surface homogeneity.

### 3.3. Availability of Functional Groups

The obtained FTIR spectrum for films prepared under the pressure levels of 0.28 t, 0.42 t and 0.84 t are shown in Figure 4 G 35 (a), G 35 (b), and G 35 (c), respectively. Usually, the available functional groups of a studied material are identified through FTIR analysis. However, since all the studied films were prepared using the same film-forming mixture (G 35), the available functional groups, and their chemical compositions were expected to be similar.

However, different interactions within molecules (i.e., starch granules and glycerol) during different film synthesis processes might have occurred under various pressure levels. Therefore, this analysis was specially conducted to investigate such changes through observing the FTIR spectrum for different prototypes of synthesised films under different hot-press conditions. The hot-press conditions, specifically the applied temperature, are reported to be affecting the characteristics of hot-pressed films [34] and, as explained before, the polymer properties are always directly and indirectly linked to the molecular level related factors (i.e., functional groups, their interactions, molecular weight, etc.). Furthermore, being a chemical reaction, the biodegradability of a particular polymer is significantly attributed by functional group availability and their interactions. Therefore, the FTIR spectrum of films synthesised under different hot-press pressure levels are worth being studied, whereas these results were also providing scientific basis to discuss the TGA results of the synthesised films.

As expected, the FTIR spectrum for G 35 (a), G 35 (b), and G 35 (c) films shown in Figure 4 are exhibiting approximately similar patterns. As reported by Choy, Prasad, Wu, Raghunandan, Yang, Phang and Ramanan [49], band assignments at around 2900 and 3300 cm^−1^ which are shown as Y and Z, respectively, in the Figure 4 are attributed to the CH_2_ stretching and O-H stretching, respectively. The peaks at around 2900 and 3300 cm^−1^ for the films suggest the presence of CH_2_ and OH groups within the studied film-forming mixture. It is well-known that each glycerol molecule (C_3_H_8_O_3_) consists of three hydroxyl groups (OH) attached to three carbon atoms arranged in a chain. The glycerol molecule also includes two CH_2_ groups. Therefore, it can be suggested that the CH_2_ stretching and the O-H stretching presented in glycerol also contributed to these peaks. Moreover, both the starch and the cellulosic compounds consisting of glucose monomers (C_6_H_12_O_6_), and their CH_2_ and OH groups might also have attributed to the emergence of these peaks. Nevertheless, the peak emerging around 3300 cm^−1^ also indicates the availability of OH groups from the absorbed water [37]. Therefore, the slight differences that can be observed around these Y and Z peaks within the prepared films might be an indicator of varied compactness and surface heterogeneity (specifically due to availability of some non-melted starch granules) that have already been discussed under the surface morphology of the films. G 35 (A) was observed to have slightly higher peak at 3300 cm^−1^ compared to other synthesised films, which might be an indicator of its comparatively high biodegradability that is attributed with high OH groups. However, it is noteworthy that the high biodegradability generally compromises the beneficial functional properties of biodegradable food packaging, and further investigations to determine the effective prototype of the synthesised films are emerging as an utmost necessity. Moreover, when comparing the three prepared films, the height of the peak at around 1620 cm^−1^ (X) for G 35 (A) is also observed to be slightly higher than that of G 35 (B) and G 35 (C). It was reported that this peak attributed to the O-H deformation of water molecules and N-H primary amine bends [50]. Therefore, while suggesting the presence of protein within the film-forming solutions, these results also suggest that the availability of water molecules within G 35 (A) is slightly higher than that of G 35 (B) and G 35 (C). Although the same film formulations and hot-pressing temperatures are being applied, these slight variations might have been influenced by the applied pressure levels during the hot-pressing process to make the bound water more available for the evaporation. Additionally, the slight variations of the O-H deformation of water molecules might also be attributed to the different water absorption capacities of three different prototypes of films synthesised through hot-pressing.

Two additional peaks were observed around 993 cm^−1^ and 1098 cm^−1^ for the FTIR spectra of the studied films. According to Bodirlau, Teaca and Spiridon [51], these peaks also indicate the presence of glycerol molecules within the film-forming solutions of different films. Therefore, the FTIR spectra for the prepared films using the hot-press confirmed the availability of glycerol molecules within the studied samples. In addition, as previously reported by Danish, Mumtaz, Fakhar and Rashid [52], a peak around the region of 1400–1420 cm^−1^ has also been observed for all the studied films, which was claimed to be a result of the bending of C-O-H groups within the glycerol molecules. Since these prepared film samples were randomly selected for this analysis, the observations suggest the ample distribution of the glycerol within the film surfaces through the different applied hot-pressed methods.

### 3.4. Thermal Stability

The obtained TGA and derivative thermogravimetric (DTG) curves for the hot-pressed G 35 (a), G 35 (b), and G 35 (c) films are shown in Figure 5a–c, respectively. The stage-wise mass loss (%), residual mass (%) and the peak temperatures (°C) of G 35 (a), G 35 (b) and G 35 (c) films are shown in Table 2. These figures present the thermal decomposition patterns of each prepared films during the TGA analysis. The obtained data was used to understand the thermal stability of the prepared films. Moreover, the obtained results for the TGA analysis will not only be used to study the thermal stability of the prepared films, but also to verify the similar film preparation conditions applied during the hot-pressing. All the films were prepared from the same film-forming mixture (G 35) at 100 °C of temperature and the only variation was the applied pressure. As expected, the obtained TGA results revealed similar thermal decomposition patterns for all the studied films, with some slight variations. According to the results (Figure 5a–c), the thermal decomposition of the studied films can be divided into four main thermal decomposition stages.

The initial thermal decomposition stage is attributed to the loss of free and absorbed water as well as the volatile compound evaporation within the studied samples [53,54]. As per the results (Table 2), G 35 (a) has reported to have a slightly higher thermal decomposition at this stage with a mass loss percentage of 5.04%. The mass loss percentages for G 35 (b) and G 35 (c) are found to be 4.00% and 4.09%, respectively. The applied high pressure during the hot-press might be attributed to the cellular level breakage of the cassava bagasse, which might facilitate internal water to become more available to the film surface. This water might be subjected to evaporation during the applied hot-press temperature resulting in less moisture availability within G 35 (b) and G 35 (c) with compared to that of G 35 (a), respectively, in Figure 5a–c. Notably, these results are also in accordance with the obtained FTIR analysis results (Figure 4) with special reference to the peak at around 1620 cm^−1^. Luchese, Benelli, Spada and Tessaro [54] have reported that the average initial mass loss percentage of different starch-based casted films were around 15%. However, these reported values are significantly different from the initial stage mass loss value of the current study. Two differently applied film-forming techniques might have attributed to this difference. These findings also provide sound explanations to support the fact that the thermal decomposition of the films might vary based on the film preparation techniques.

The second main thermal decomposition of all the studied films has been found to be commenced at a temperature range of 183–185 °C, and has generally lasted until 250 °C. According to the obtained results (Figure 5a–c), this stage was found to be the most prominent thermal decomposition stage for all the studied films. Luchese, Benelli, Spada and Tessaro [54] reported an intermediary stage during the thermal decomposition, which occurred at temperatures above the boiling point of glycerol (182 °C). A shoulder was also reported to be appeared within the temperature range of 240–260 °C, which was attributed to the low-molecular-weight reaction products. However, the most prominent decomposition stage reported by Luchese, Benelli, Spada and Tessaro [54] occurred within the temperature range of 200–350 °C. They also reported that this is attributed to the decomposition of the glycerol-rich phase including the starch decomposition. Moreover, the findings of Edhirej, Sapuan, Jawaid and Ismarrubie Zahari [53] confirm that the mass loss around the temperature of 220 °C is mainly attributed to the volatilisation of plasticisers along with some water content. Although the temperature ranges are varied, it is proven that the mass loss at the second stage (Figure 5a–c) is mainly attributed to the decomposition of glycerol molecules. Accordingly, G 35 (c) has reported to have the highest mass loss (40.63 %) during the second thermal decomposition stage, whereas the mass losses for G 35 (a) and G 35 (b) account for 40.50% and 37.73%, respectively (Table 2). Since same film-forming mixture (G 35) has been used for the film preparation, these slight variations suggest the possible differences in the adhesion of glycerol molecules to the cassava bagasse matrix. Accordingly, though all the studied films are exhibiting approximately similar adhesion among glycerol and matrix, comparatively high adhesiveness for G 35 (b) was observed. The obtained SEM micrographs (Figure 3) also support this claim, in which less defects and features on the film surfaces are observed. The only possible reason for this variation is the applied pressure on the film-forming mixtures during the hot-pressing. Therefore, it can be suggested that the applied 0.42 t for the film-preparation might be more appropriate to maintain thermal stability within the films. This claim also can be supported by the obtained mass loss percentages at the initial decomposition stage.

However, more experimentations in future on thermal stability and the pressure conditions for the film-forming mixture (G 35) at 100 °C is recommended to establish the findings. Furthermore, the DTG graphs (Figure 5a–c) for the current study have reported a prominent peak around 220 °C for all the studied samples. Guimarães, Botaro, Novack, Teixeira and Tonoli [55] also reported a similar prominent peak at around 248 °C for the obtained DTG curve for the thermal decomposition of the pristine polyvinyl alcohol (PVA). This suggests an impressive thermal stability of the utilised G 35 film-forming mixture and its resulted films, even with the absence of any modifications to the film-forming mixtures. Moreover, these findings suggest the feasibility of enhancing the thermal stability of the prepared films through various modifications to the film-forming mixture.

The third major thermal decomposition event for all the studied films has been reported around 250–390 °C. According to the DTG curves, all these studied films have reported the peak around 310 °C. Although this was reported to be the maximum decomposition temperature for previously studied starch-based films [53,54,55], this was contrary to the findings of the current study. Instead, the peak at the DTG curves (Figure 5b) has been found to be the second-highest peak. According to Aggarwal, Dollimore and Heon [56], the mass losses at temperatures higher than 250 °C are attributed to the breakdown of cellulose and starches. Therefore, as shown in Table 2 the reported mass losses for G 35 (a), G 35 (b), and G 35 (c), accounted for 25.11%, 24.90%, and 26.08%, respectively, reflect the initial thermal decomposition of starch and cellulosic substances within the cassava bagasse matrix. Accordingly, the mass losses at the third thermal decomposition stage for G 35 (a) and G 35 (b) are approximately similar to each other, whereas the G 35 (c) exhibits comparatively high thermal decomposition for all the stages.

The last thermal decomposition stage occurred at temperatures over 390 °C, and this was attributed to the carbonisation process of the films [55]. As shown in Table 2, the reported mass losses for the G 35 (a), G 35 (b), and G 35 (c) during this stage were 9.23%, 12.02%, and 12.37%, respectively. Accordingly, unlike during the other decomposition stages, for temperatures higher than 400 °C, G 35 (a) has exhibited the highest thermal stability. However, the residual masses for G 35 (a), G 35 (b), and G 35 (c) were found to be 14.24%, 15.44%, and 10.65%, respectively (Table 2). It has been reported that the starch possesses a natural charring ability due to its polyhydric nature [57]. The cassava bagasse used to synthesise the films consists of a reasonable amount of residual starch, which might have also contributed to the formation of char. Moreover, higher residual mass indicates higher amount of stabilised char formation [57,58]. Thereby the residual mass after the thermal decomposition of the studies films may indicate the efficiency of the plasticisation of starch granules with glycerol while resulting in more homogeneous and compatible films. Accordingly, the highest residual mass has been observed during the thermal decomposition of G 35 B (Table 2), and the surface morphology of the same film was also reported to be impressive. Moreover, as explained by Edhirej, Sapuan, Jawaid and Ismarrubie Zahari [53], G 35 (c) (Figure 5c) exhibits comparatively less thermal stability, whereas the thermal stability of both the G 35 (a) (Figure 5a) and G 35 (b) (Figure 5b) is approximately similar. Overall, based on the above analysis for each stage it can be claimed that the thermal stability of G 35 (b) (Figure 5b) for temperatures below 400 °C is slightly higher than that of G 35 (a) (Figure 5c).

## 4. Conclusions

The current study investigated the potential of direct utilisation of cassava bagasse as an alternative, non-edible, starch-based matrix to synthesise biodegradable food packaging through hot-press method followed by preliminary characterisations (i.e., morphological, chemical, and thermal properties). The optimum glycerol content (as plasticiser), hot-pressing temperature and the applied pressure to synthesise the prototype of cassava bagasse-based films were found to be 35%, 100 °C, and 0.42 t, respectively. Despite few non-melted starch granules and defects, the SEM micrographs indicated impressive film surface morphologies, whereas satisfactory distribution and compatibility between residual cassava bagasse starch granules and glycerol were also suggested. The FTIR analysis results confirmed the approximately similar chemical compositions within all studied prototypes of the film samples. The observed slight variations suggesting slight changes to their functional properties and biodegradability. Despite the slightly high thermal stability of G 35 (B), G 35 (A) and (B) have exhibited similar thermal stabilities. Overall, the feasibility of the direct utilisation of cassava bagasse as an alternative, non-edible starch-based matrix for film synthesis has been proven. However, further investigations (i.e., mechanical and functional properties, food integrity, biodegradability) are required to determine the appropriateness of the synthesised films for developing commercial-scale biodegradable food packaging. As future research insights, the potentials film extrusion, 3D printing of the films and comprehensive investigations on property enhancement without compromising the biodegradability of the synthesised prototypes of the films are recommended. Successive outcomes of such further investigations are expected to promote the valorisation of agro-industrial waste, environmental sustainability, and food security in the long run.

## Figures and Tables

**Figure 1 polymers-15-00767-f001:**
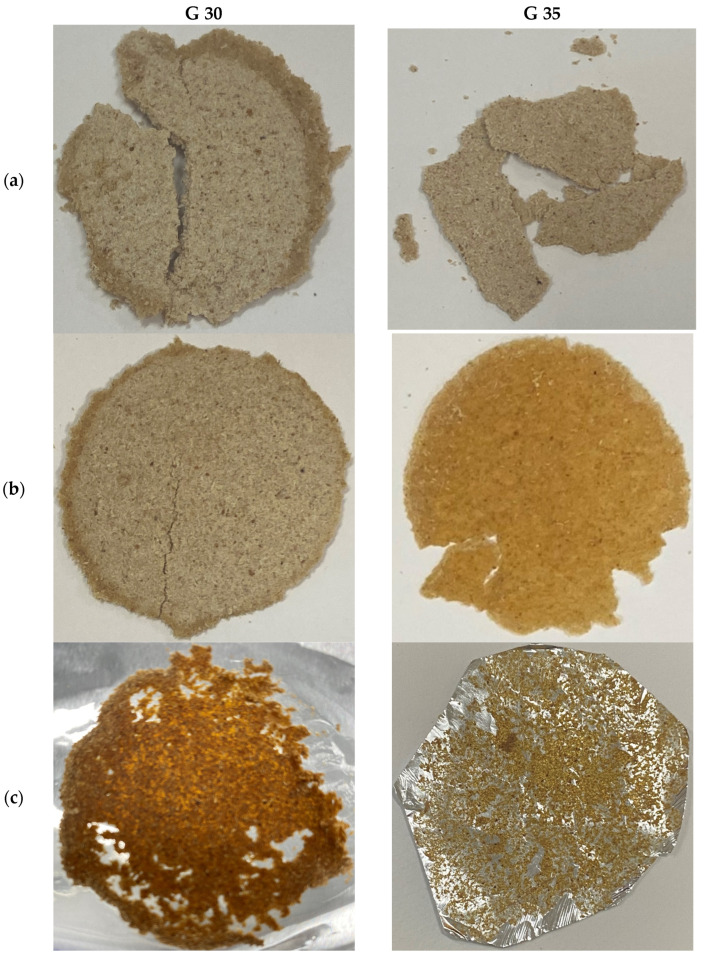
Macroscopic views of hot-pressed films prepared using G 30 (30% glycerol) and G 35 (35% glycerol) film-forming mixtures with (**a**) 60 °C, (**b**) 100 °C and (**c**) 140 °C temperature levels at 0.28 t pressure level.

**Figure 2 polymers-15-00767-f002:**
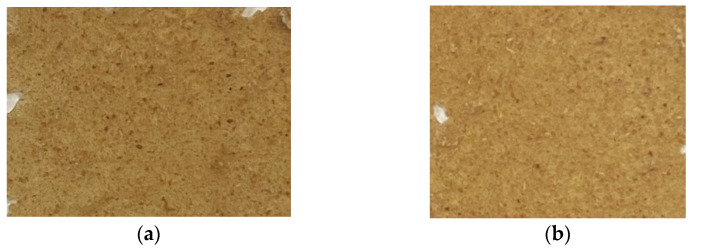
Macroscopic views of hot-pressed films prepared using G 35 (35% glycerol) film-forming mixtures by applying (**a**) 0.42 t and (**b**) 0.84 t pressure levels.

**Figure 3 polymers-15-00767-f003:**
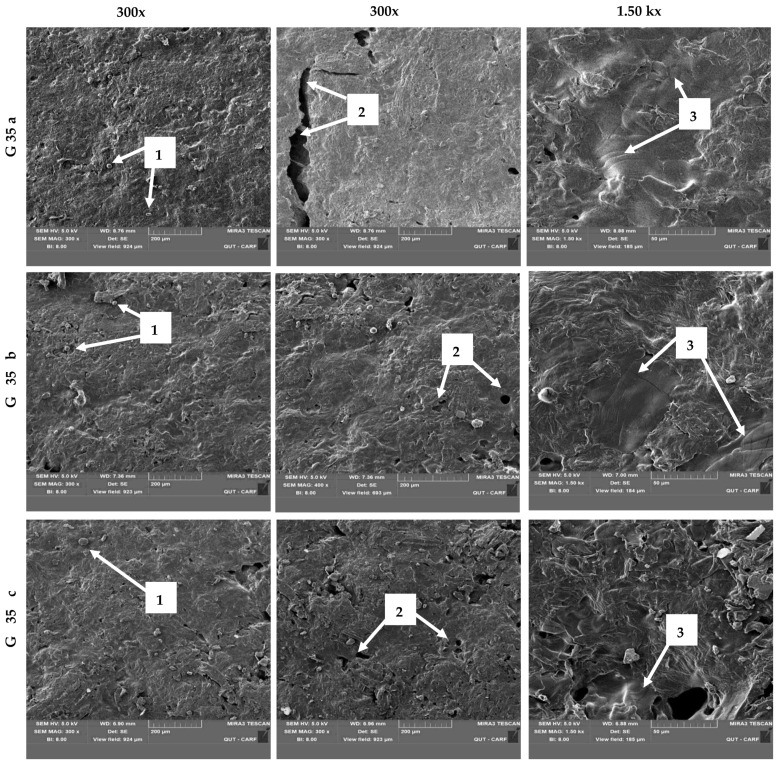
Scanning electron microscopic images of the hot-pressed films prepared with G 35 (35% glycerol) film-forming mixture under (**a**) 0.28 t (**b**) 0.42 t and (**c**) 0.84 t at 100 °C temperature observed under 300 x and 1.5 kx magnification levels.

**Figure 4 polymers-15-00767-f004:**
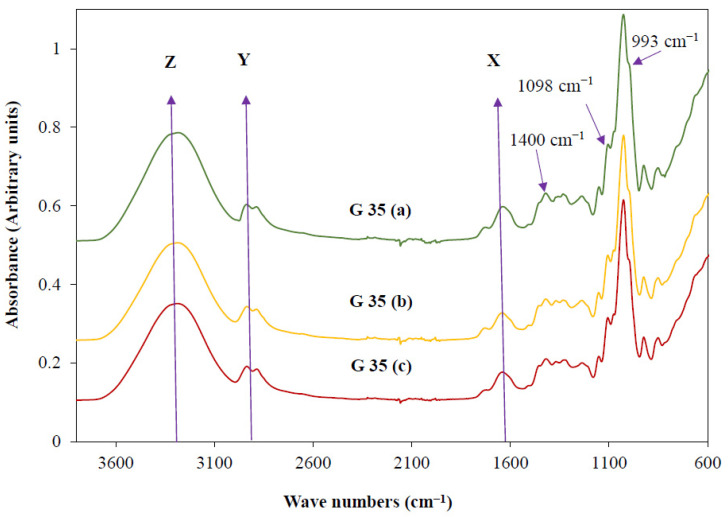
Fourier Transform Infrared (FTIR) spectrum for hot-pressed film prepared using 35% glycerol (G 35) with a pressure of 0.28 t (G 35 (**a**)), 0.42 t (G 35 (**b**)), and 0.84 t (G 35 (**c**)).

**Figure 5 polymers-15-00767-f005:**
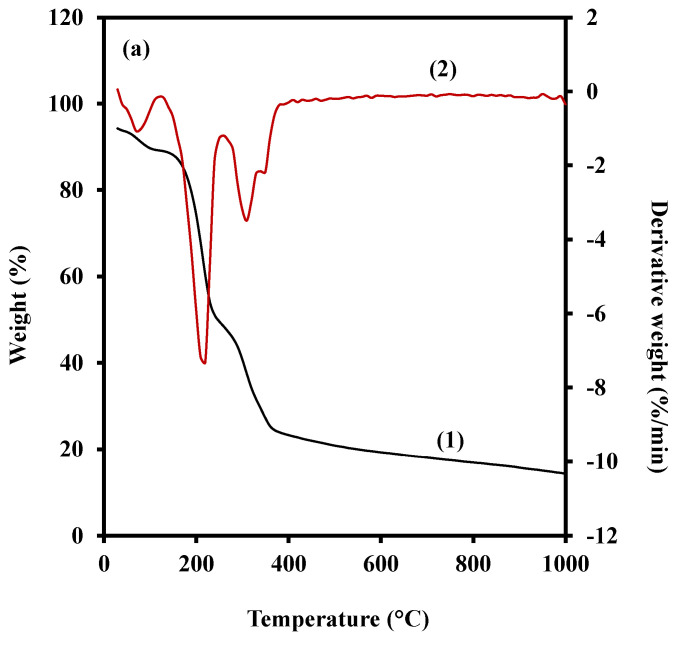
(1) The thermogravimetric analysis (TGA) curves and (2) derivative thermogravimetric (DTG) curves for G 35 (**a**), G 35 (**b**), and G 35 (**c**) films prepared using 0.28 t, 0.42 t, and 0.84 t pressure levels, respectively.

**Table 1 polymers-15-00767-t001:** Formulations for film-forming mixtures for the hot-press method.

Film	Powdered Cassava Bagasse ^1^	Glycerol ^1^	Water ^1^	Glycerol + Water
G 30	47	30	23	53
G 35	37	35	28	63

^1^ The constituents are reported in their weight percentages.

**Table 2 polymers-15-00767-t002:** Stage-wise mass loss (%), Residual mass (%), and peak temperatures (°C) of G 35 A, G 35 B and G 35 C films.

Film	Stage 1			Stage 2			Stage 3			Stage 4		
	Mass Loss	Residual Mass	Peak Temper-Ature	Mass Loss	Residual Mass	Peak Temper-Ature	Mass Loss	Residual Mass	Peak Temper-Ature	Mass Loss	Residual Mass	Peak Temper-Ature
G 35 A	5.04	89.10	97.6	40.50	48.60	178.6	25.11	23.48	355.2	9.23	14.24	567.1
G 35 B	4.00	90.09	91.9	37.73	52.36	179.1	24.90	27.46	348.3	12.02	15.44	993.6
G 35 C	4.09	89.73	30.6	40.63	49.10	175.9	26.08	23.03	352.8	12.37	10.65	995.2

Mass loss and residual mass are presented in their percentage values (%), while the onset and peak temperatures are given in °C.

## Data Availability

The data presented in this manuscript are available on request from the corresponding author.

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
