# Peer review of "Investigation of Morphological, Chemical, and Thermal Properties of Biodegradable Food Packaging Films Synthesised by Direct Utilisation of Cassava (*Monihot esculanta*) Bagasse"

_polymers, 2023, doi:10.3390/polym15030767_

Round 1
Reviewer 1 Report
The manuscript titled “Investigation of morphological, chemical, and thermal properties of biodegradable food packaging films synthesised by direct utilisation of cassava (Monihot esculanta) bagasse” is aimed to synthesis and characterise a cassava bagasse-based biodegradable food packaging material in order to assess the feasibility of cassava bagasse as an alternative matrix. The authors expected that the synthesized films will exhibit full biodegradability, and also will assist ensuring the food security during their large-scale commercial applications, and tried to prove it during the experiment. Generally, the topic is relevant and the work has scientific novelty. Therefore, the manuscript deserves consideration. However, in current state the presented work needs revision. The main issues are listed below.
1. Introduction section should be modified:
1.1. L. 44-46. “Additionally, food packaging delivers the information on the packed food to the consumers and also indirectly caters as a powerful marketing tool by influencing consumer behaviour [3]”. Consumer behavior is very important aspect for each study on sustainable food packaging and new biodegradable materials. The authors should mention consumer’s opinion and behavior in several sentences to justify the topic relevance. Study can be interesting but not important if the final product is not interesting to consumers. Also I found cited source [3] is not full suitable for consumer behavior consideration. It is a well written comprehensive review, but consumer’s behavior mentioned only in Introduction and very general. Alternative I suggest to consider the next sources on consumer’s behavior towards new trends in food packaging:
https://doi.org/10.1016/j.fufo.2022.100191
https://doi.org/10.3390/su13031463
https://doi.org/10.3389/fsufs.2021.630393
1.2. L. 62-65. This part also should be expanded. Before moving on to discussing the benefits of starch other interesting materials should be mentioned. There are many well designed works on development of biodegradable films based on gelatin, chitosan, modifications of ethyl-, methylcellulose. For consideration:
https://doi.org/10.1038/s41598-022-16878-w
https://doi.org/10.1038/s41598-022-20751-1
https://doi.org/10.1038/s41538-022-00132-8
2. In the manuscript, the authors indicated that the work is aimed at both synthesis and investigation of the characteristics of a biodegradable film. In the Introduction, the authors pointed out that “Any organic material that has this ability can be defined as biodegradable materials. (L. 61-62).“ However, this is not enough to claim the biodegradable properties of the developed films. Throughout the entire work, there are no studies of the main characteristics of the film, such as tensile strength, adhesion coefficients and water absorption. Focusing on the possibility of biodegradation, it is necessary at least to provide data on solubility in water and decomposition time.
3. Figure 1 shows photographs of the obtained film samples. How are the authors going to use the product being developed in the food industry when it does not have the necessary strength, transparency and other properties? Did the authors compare their film with analogues? Such a product is no longer like a film, but an eco-paper. To avoid confusions, I recommend to remove the Figure 1 or move it to Supplementary.
4. Figure 4. How the authors try to explain the influence of pressure on the change in structure and the presence of fluctuations in the bonds of functional groups in the samples. The pressure change is insignificant, therefore, the study by IR spectroscopy is not of a significant scientific nature. The fact is that the change of bands in the IR spectra begins to manifest itself with an increase in pressure by tens, hundreds of times. I recommend the authors to expand the use of IR spectroscopy in the study of their samples in this work (if possible) and in subsequent works.
5. It is necessary to present a visual diagram or in a formulaic form the temperature transformations described in the study of samples by the method of thermogravimtery. Also, this subsection does not describe the processes that occur when the temperature rises from 500 OC and above. What substances remain there?
Based on points above I recommend to reconsider the manuscript after major revision.
Author Response
Please find the responses in the attached file.

Reviewer 2 Report
The article entitled “Investigation of morphological, chemical, and thermal properties of biodegradable food packaging films synthesised by direct utilisation of cassava (Monihot esculanta) bagasse”. The authors developed an efficient way to produce cassava starch-based films. The article is interesting and the experiments are accurately described. I suggest this article can be accepted after minor modifications.
1. Line 118: The compositions of cassava bagasse should be provided.
2. Line 334: Will these non-melted starch granules affect the functional properties of the films?
3. Fig.4: The intensity of peaks at around 2900 and 3300 cm-1is also deferent among samples.
4. It would be better if the authors could provide more functional properties, such as mechanical properties and barrier properties.
Author Response

(The authors gave the same response as above.)

Reviewer 3 Report
Although authors did quality work, however with the limited experiments the manuscript is incomplete. A major revision is recommended and invited to submit the revised version.
Suggested to perform some major experiments including physical characterization of film, mechanical property, zeta potential, biocompatibility, contact angle, vapor permeability, and specially perform food packaging applications to prove the applicability and suitability of the film. Moreover, as indicated in title biodegradable polymer, suggested to perform degradability test.
Line no 22. authors said very limited study available, A significant study available on cassava bagasse. suggested to rewrite the abstract in structured format
The films prepared demonstrated impressive mor- 30 morphological characteristics. Suggested to provide impressive data
The authors are suggested to combine paragraph 2 and 3 of introduction and focus on related draft as not significantly correlating with title
What was the relevant hypothesis behind selection of high concentration of plasticiser
Line no. 151-152: on what basis the pressure (0.28 t ) was selected and further what was the hypothesis of selecting the pressure (0.42 t and 0.84 t ). did author tested any other pressure, if so suggested to provide details as supplementary data.
At what temperature and relative humidity the film were stored.
The author wrote Figure 1 is macroscopic view, please explain in detail in methodology section
Please check the value of FTIR as mentioned at line no. 187 does not match with the figure, the analysis said performed in between 400 - 4000.
Line no 424. the wave number presented are missing in the figure
Authors suggested to provide a tabular data indicating stage wise loss, exothermic or endothermic temperature and percentage loss with residuals from thermogravimetric results.
Authors are suggested to write concise conclusions not summary.
Some minor corrections are also reflected in the attached file.

Author Response

(The authors gave the same response as above.)

Reviewer 4 Report
This work reported the potential of direct utilization of cassava bagasse waste-based to be used as both starch and fiber-rich alternative matrices to synthesize eco-friendly biodegradable food packaging materials and their characterizations. These attempts at direct utilization of cassava bagasse as a fiber starch matrix without extracting starch for film preparation would avoid any constraints or complications associated with its starch extraction process. Prepared two film-forming mixtures by incorporating glycerol (30 % and 35%), powdered cassava bagasse, and water. By analyzing the morphology, functional group availability, and thermal stability of prepared films, the promising condition for film preparation can be obtained that application of 35% glycerol, with 100 °C, 0.42t temperature, and pressure respectively. The films prepared demonstrated impressive morphological characteristics. The biodegradable food packaging films would ensure better food security and helpful in achieving economic benefits through their valorization. Overall, this work can be published after addressing the issues below.
1. In Figure 1, why do these samples have contrast in colors?
2. How to characterize the flexibility of G35b?
3. Whether the presence of sugars at the high-temperature process affect the quality of final films?
4. Some important works should be cited, such as Chemical Engineering Journal 429 (2022) 132342; ACS Appl. Mater. Interfaces 2022, 14, 11672−11680
5. Are the starch granules the direct reason causing the defects in the surface of the films?
6. Can these prepared films be tailored into specific shapes for practical application?
Author Response

(The authors gave the same response as above.)

Round 2
Reviewer 1 Report
The authors carefully considered all comments and gave detailed response to each comment. I find revised version suitable for publication in Polymers and would like to recommend to accept this work.
Reviewer 3 Report
Authors have reflected all the suggestions and corrections, the manuscript can be accepted in present form.